# Clinical Utility of *LCT* Genotyping in Children with Suspected Functional Gastrointestinal Disorder

**DOI:** 10.3390/nu12103017

**Published:** 2020-10-01

**Authors:** María L. Couce, Paula Sánchez-Pintos, Emiliano González-Vioque, Rosaura Leis

**Affiliations:** 1Department of Pediatrics, University Clinical Hospital of Santiago de Compostela, IDIS-Health Research Institute of Santiago de Compostela, 15704 Santiago de Compostela, Spain; emiliano.gonzalez.vioque@sergas.es (E.G.-V.); mariarosaura.leis@usc.es (R.L.); 2CIBERER, Instituto Salud Carlos III, 28029 Madrid, Spain; 3Department of Pediatrics, Universidad de Santiago de Compostela, 15704 Santiago de Compostela, Spain; 4CIBEROBN, Instituto Salud Carlos III, 28029 Madrid, Spain

**Keywords:** lactose intolerance, lactose malabsorption, C/T-13910 polymorphism, functional gastrointestinal disorders

## Abstract

Genetic testing is a good predictor of lactase persistence (LP) in specific populations but its clinical utility in children is less clear. We assessed the role of lactose malabsorption in functional gastrointestinal disorders (FGID) in children and the correlation between the lactase non-persistence (LNP) genotype and phenotype, based on exhaled hydrogen and gastrointestinal symptoms, during a hydrogen breath test (HBT). We also evaluate dairy consumption in this sample. We conducted a 10-year cross-sectional study in a cohort of 493 children with suspected FGID defined by Roma IV criteria. Distribution of the C/T-13910 genotype was as follows: CC, 46.0%; TT, 14.4% (LP allele frequency, 34.1%). The phenotype frequencies of lactose malabsorption and intolerance were 36.3% and 41.5%, respectively. We observed a strong correlation between genotype and both lactose malabsorption (Cramér’s V, 0.28) and intolerance (Cramér’s V, 0.54). The frequency of the LNP genotype (*p* = 0.002) and of malabsorption and intolerance increased with age (*p* = 0.001 and 0.002, respectively). In 61% of children, evaluated dairy consumption was less than recommended. No association was observed between dairy intake and diagnosis. In conclusion, we found a significant correlation between genotype and phenotype, greater in older children, suggesting that the clinical value of genetic testing increases with age.

## 1. Introduction

The digestion of lactose, a common disaccharide in human nutrition, is dependent on intestinal lactase (LCT) activity. This enzyme, also known as lactase-phlorizin hydrolase, is a β-d-galactosidase found in the apical surface of the intestinal microvilli. LCT expression begins before birth, remains high during the nursing period, and then, in most people, progressively declines after weaning, resulting in lactase nonpersistence (LNP) [1]. This genetically programmed reduction in lactase activity varies greatly between populations, with lowest levels observed in Nordic populations (<5% in Denmark) and highest in Korean and Han Chinese populations (approaching 100%). Large variations are also observed at the regional level [2], affecting about 70% of the adult population globally. Positive selection of lactase persistence (LP) is usually explained by the gene-culture coevolution hypothesis, whereby LP confers a nutritional advantage in pastoral populations [3,4], and by the calcium assimilation hypothesis, which proposes a selective benefit of LP in environments with low ultraviolet irradiation and low dietary vitamin D intake (e.g., high-latitude regions) [5]. LP is common in people of European ancestry, especially in northwestern Europe, and declines in frequency further south and east.

In Caucasians, differences in lactase activity have been linked to two single-nucleotide polymorphisms (SNPs), C/T−13910 and G/A−22018, both of which are located upstream of the LCT-encoding gene [6]. Both variants are in strong linkage disequilibrium, and functional evidences indicates that C/T−13910 variant is responsible for LP. In Caucasians, the CC and GG variants are associated with hypolactasia and are good predictors of the decrease in intestinal lactase expression, whereas TT and AA_-_ genotypes are predictors of LP. Genetic LP is considered a dominant genotype and CT and GA carriers present intermediate levels of lactase expression. LNP results from the accumulation of transcriptionally suppressive epigenetic changes in haplotypes carrying the SNP C (−13,010) allele, whereas in T (−13,910)-containing haplotypes epigenetic inactivation is avoided, facilitating LP [7]. The distribution of these different lactase phenotypes in human populations is highly variable [8]. Spanish series have reported frequencies of the LCT persistence allele ranging from 36.8% to 66% [9,10,11,12].

LP is one of the most influential factors affecting human dietary patterns [13], and it has been associated with several potential risks in adults, including a higher body mass index (BMI) in European LP populations [11,14,15], especially those that consume large amounts of dairy products [11,16], and metabolic syndrome [10]. Although findings are conflicting, LP is also proposed to influence bone health and fracture risk [17,18,19].

Several reports [20,21,22,23,24] have evaluated the clinical value of genetic testing, which can be a good predictor of LP/LNP in specific populations. A high correlation between LP and the standard hydrogen breath test (HBT) has also been demonstrated [21,24,25], even though the utility of this test may be limited by ethnicity [23]. Concordance between these two diagnostic tools appears to be influenced by age [26], with greater correlation observed in those over 30 years [20]. The pediatric clinical value of genetic testing and its correlation with HBT findings in childhood requires further evaluation, since most comparative genotyping studies have been performed in adult populations.

In this study, we assessed the role of lactose malabsorption in functional gastrointestinal disorders (FGID) in children, and examined the correlation between the LNP genotype and phenotype, defined based on exhaled hydrogen (H2) measurement and gastrointestinal (GI) symptoms during HBT, in a pediatric population.

## 2. Materials and Methods

### 2.1. Study Design and Population

A cross-sectional, observational, single-center study was performed at the Pediatric Gastroenterology, Hepatology and Nutrition Unit of the University Clinical Hospital of Santiago de Compostela, Spain. The study was approved by the Local Ethics Committee of Santiago-Lugo (code: 2020/319). All procedures were conducted in accordance with the Declaration of Helsinki and signed informed consent was obtained from parents and from children aged >12 years.

Between 1 January 2010, and 31 December 2019, 493 consecutive pediatric patients (sample A) with GI symptoms compatible with FGID, defined according Rome IV criteria [27], were evaluated. Exclusion criteria included age > 18 years and the presence of major metabolic diseases. The study population was subdivided into three age groups: ≤5 years; 6–11 years; and ≥12 years. The following variables were recorded at recruitment: age; gender; anthropometric data; family history of lactose intolerance; symptoms for which patient consulted; exhaled H2 and GI symptoms during HBT; and genetic study of SNP C/T-13910.

Malabsorption was defined according to HBT test results as a H2 increase ≥ 20 ppm with respect to baseline, and intolerance was defined as the presence of symptoms during the HBT test.

In a subgroup of 120 children (sample B), we also analyzed G/A−22018 polymorphisms and serum levels of calcium (Ca), phosphorous (P), parathormone (PTH), calcitonin, calcitriol, and 25-OH vitamin D (25-OHD), and used a validated questionnaire to collect data on dairy consumption, including the number of weekly milk, cheese, and yogurt servings and the weekly quantity of milk consumed.

All biochemical measurements were obtained from fasting morning plasma samples at the same time of day (8.00 h) and all patients were free of acute infection and not receiving medication. Reference intervals for biochemical parameters are as follows: Ca: 8.6–10.3 mg/dL (ages 1–2 years), 9.2–10.3 mg/dL (ages 3–6 years), 8.7–10.5 mg/dL (males aged 6–9 years), 9.0–10.6 mg/dL (females aged 6–9 years), 9.0–10.5 mg/dL (ages 10–14 years), 8.8–10.5 mg/dL (ages 15–18 years); P: 3.1–5.6 mg/dL; PTH: 12–72 pg/mL; calcitonin: 0–18.2 pg/mL (males), 0–11.5 pg/mL (females); calcitriol: 20–54 pg/mL. 25-OHD levels are considered normal at >20 ng/mL, while levels of 12–20 ng/mL and ≤12 ng/mL are considered indicative of 25-OHD insufficiency and deficiency, respectively [28]. 

### 2.2. Anthropometric Measurements

Standing height was measured with a wall-mounted stadiometer and body weight, to the nearest 100 g, with digital scales. Patients were weighed barefoot after overnight fasting. BMI was calculated as weight (kg)/height^2^ (m^2^). Subjects were classified according to BMI by using WHO Child Growth Standards for underweight (BMI percentile, <5), normal weight (BMI percentile, 5–<85), overweight (BMI percentile, 85–94), and obese (BMI percentile, ≥95) [29].

Percentiles and z-scores for anthropometric measurements were calculated using the online nutritional assessment tool of the Spanish Society of Gastroenterology, Hepatology, and Nutrition (https://www.seghnp.org/nutricional/) [30].

### 2.3. Analytical Measurements

Concentrations of Ca and P were determined by standard procedures with the Advia 2400 Analyzer (Siemens Healthcare Diagnostic Systems, Erlangen, Germany); 25-OHD with the Advia Centaur XP Analyzer (Siemens Healthcare Diagnostics, Erlangen, Germany); PTH with Roche Cobas E 601 (Roche Diagnostic, Indianapolis, IN, USA); calcitonin with INMULITE 2000 (Siemens Healthcare Diagnostics, Erlangen, Germany); and calcitriol by radioimmunoassay in Reference Laboratory S.A. (Hospitalet de Llobregat, Spain).

### 2.4. Hydrogen Breath Test

The HBT test consists of measurement of the concentration of H2 in exhaled air after 12 h of fasting and administration of lactose (2 g/kg up to a maximum of 50 g) diluted in aqueous solution (20%). Standard requirements for the HBT were applied: absence of treatment in the preceding 15 days with antibiotics, pre/probiotics, or laxatives, or any other drug that could alter the colonic flora; avoidance of fiber-rich food the preceding 3 days; avoidance of physical activity before or during the test; and no smoking prior to testing.

A sample is obtained upon exhaling through the mouthpiece after breathing deeply. The concentration of hydrogen in the exhaled air sample is measured using a gas chromatograph (CM2 MicroLyzer, Quin Tron, Milwaukee, WI, USA). Measurements were taken immediately before (0 min, basal measurement) lactose administration and afterwards at 30-min intervals for 3 h (30, 60, 90, 120, 150, and 180 min). Peak H_2_ was recorded.

The results of the HBT were interpreted as follows: absorption (increase <10 ppm H_2_ with respect to baseline); poor absorption (increase ≥10 ppm and <20 ppm H_2_ with respect to baseline); malabsorption (increase ≥20 ppm H_2_ with respect to baseline). Exclusion criteria included basal H_2_ levels > 30 ppm. If basal H_2_ levels were >10 ppm and <30 ppm, we checked that the measurement at 30 min decrease.

Besides, any GI symptom (abdominal pain, nausea, vomiting, diarrhea) reported by subjects during the test between 0 and 180 min were recorded.

Tolerance was defined as the absence of symptoms during the test and intolerance as the presence of symptoms during the test.

### 2.5. Genetic Study

Genotyping of C/T−13910 and G/A−22018 SNPs was performed using the SEQUENOM Platform (Agena Bioscience, San Diego, CA, USA).

### 2.6. Statistical Analysis

To verify the homogeneity of distribution of quantitative variables, we used the Student’s *t*-test for normally distributed variables, the non-parametric Mann–Whitney U test for variables with a non-normal distribution, and the Kruskal–Wallis test for comparisons between more than 2 groups. Normality of the variables was evaluated using the Shapiro–Wilk statistical test and homoscedasticity using the Bartlett test. In cases in which outliers were detected in the data or the conditions of homoscedasticity were not met, we employed the robust generalization of the Welch test or the Yuen test, using the trimmed mean as an estimator with a cut-off level of 0.2. For analysis of qualitative variables, we used the Chi-square test or Fisher’s exact test in cases of non-compliance with the assumptions of the Chi square test.

The degree of association between variables was assessed using Cramér’s V test, which returns values between 0 and 1, where 0 indicates an absence of a relationship and 1 a perfect relationship. Usually accepted cut-off points are as follows: 0.1, weak relationship; 0.3, median relationship; and 0.5, strong relationship.

The agreement between HBT results and those of lactase genotyping was assessed using Cohen’s Kappa coefficient. Analyses were performed using R Core Team (version 3.6.3, 2020; R Foundation for Statistical Computing, Vienna, Austria). *p*-values obtained were adjusted using the Bonferroni correction. Only adjusted *p*-values < 0.05 were considered statistically significant.

## 3. Results

### 3.1. Characteristics of the Study Population

The study population (*n* = 493) (sample A) showed a homogeneous distribution in terms of gender (234 males, 47.4%). Age distribution was as follows: ≤5 years, *n* = 50 (10.1%); 6–11 years, *n* = 326 (66.12%); ≥12 years, *n* = 117 (23.7%). The characteristics of the study population are summarized in Table 1. The predominant symptom reported on consultation were abdominal pain, diarrhea, and nausea, and the frequency of abdominal pain (*p* = 0.008) and nausea (*p* = 0.009) increased significantly with age.

The C/T-13910 genotype distribution was as follows: CC, 46.0%; CT, 39.5%; TT 14.4%. The frequency of the LP allele was 34.1%. A significant increase in LNP genotype was observed with increasing age (*p* = 0.002).

### 3.2. Phenotype

The mean frequencies of malabsorption and lactose intolerance in our series were 36.3% and 41.5%, respectively. With increasing age, the frequency of both parameters increased significantly and progressively (malabsorption, *p* = 2.2 × 10^−16^; lactose intolerance, *p* = 0.003). In line with this observation, peak H_2_ and H_2_ increase also increased with age (*p* < 0.001) (Table 1). 

As reflected in Table 2, despite the higher values obtained for peak and H_2_ increase in individuals with lactose malabsorption, we found a clear difference for both markers between lactose-intolerant and lactose-tolerant children, with significantly higher values observed in the intolerant group (*p* < 0.001). In lactose-tolerant children aged >5 years, we observed a progressive age-associated increase in time to peak H_2_ (*p* = 0.005). Time to peak H_2_ was also higher in lactose-intolerant than in lactose-tolerant children. The most frequently observed symptom induced by the HBT was abdominal pain, followed by flatulence, diarrhea, and nausea.

### 3.3. Phenotype/Genotype Correlation

We observed moderate agreement between HBT and the results of genetic testing (Cohen’s Kappa, 0.55; 95% CI, 0.49–0.61). As shown in Table 3, C/T-13910 polymorphism was significantly correlated with phenotype of absortion/malabsorption by HBT (*p* < 2.22 × 10^−16^) and of lactose tolerance/intolerance (*p* = 2.258 × 10^−09^). Evaluation of the strength of those relationships revealed strong association for malabsorption (Cramér’s V, 0.54), and a less strong but non-negligible association for intolerance (Cramér’s V, 0.28). It should be noted that 98.8% of the children with lactose malabsorption were C-allele carriers, and 95.5% were homozygous for the C-allele. Nonetheless, 215 C-allele carriers were identified as lactose absorbers based on HBT results. For both CC and CT genotypes, the frequency of lactose absorbers decreased with increasing age (Table 3).

We developed a post-hoc test to examine differences in lactose absorption and lactose tolerance according to C/T-13910 genotype (Figure 1). We observed significant differences in lactose absorption between the CC genotype and the other two genotypes, and in lactose tolerance between each of the three C/T-13190 genotypes analyzed (Appendix A).

In sample B, in which we analyzed G/A-22018 polymorphisms, all children with lactose malabsorption were G-allele carriers, and we observed significant differences in lactose malabsorption between the GG genotype and the other two genotypes analyzed. Moreover, there was a strong relationship between C/T-13190 and G/A-22018 polymorphisms (*p* = 2.2 × 10^−16^; Cramér’s V value, 0.96) (Appendix A).

### 3.4. Dairy Product Intake and Phosphocalcic Metabolism

As shown in Table 4, evaluation of 120 children using the dairy intake questionnaire revealed that 10.9% of the study population did not consume milk daily, 15.9% did not consume yoghurt daily; and 28.4% did not consume cheese daily. The majority of children, in all age groups, consumed 5–7 servings of milk per week. However, estimated milk consumption varied widely, with a mean weekly consumption of 2604.58 ± 1477.60 mL. We observed no significant differences in dairy consumption according to age (*p* = 0.46) or polymorphism (*p* = 0.69). The analysis of bone health biomarkers revealed no significant differences between groups with lactose malabsorption, tolerance, and intolerance or between genotypes (data not shown).

## 4. Discussion

This study of a population of children with FGID symptoms, defined according to Rome IV criteria, reveals a prevalence of lactose malabsorption similar to that of the general population, a high correlation between phenotype and genotype, and low dairy consumption.

Our results show that a diagnosis of lactose malabsorption (defined based on HBT results) was established in about one third (36.3%) of children with recurrent abdominal pain due to suspected functional disorder. This frequency is similar to that reported in other series in children with chronic abdominal pain [31], and in line with that observed in a healthy population in Spain [32]. Moreover, the frequency of malabsorption increased with age, from 8% in children ≤5 years to 49.5% in children aged >12 years (*p* = 0.001). Lactose malabsorption therefore appears not to play a major role in FGID in younger children, in agreement with previous studies indicating that carbohydrate malabsorption may be an incidental finding in children with FGID, rather than its cause [31]. When evaluating FGID in infants and children, we should screen for celiac disease (CD) as its prevalence among children with irritable bowel syndrome is higher than in general pediatric population [33]. It should also be borne in mind that allergy to cow’s milk protein and CD may act as a predisposing or coexisting factor, potentially contributing to inflammation and visceral hypersensitivity in early life, which may manifest as FGID [34,35].

Although several diagnostic tests are available for malabsorption, diagnosis of lactose intolerance remains a challenge [36]. In this study, intolerance was defined based on the presence of symptoms during HBT. Because genetic, enzyme activity, and breath tests only reveal enzyme deficiency, lactose maldigestion, or lactose malabsorption, validated symptom assessment is required to diagnose intolerance [37]. The clinical value of LCT genotype is based on the fact that is a good predictor of LP/LNP in specific populations and shows a good correlation with HBT results. In Caucasians, LP is almost uniformly mediated by the C/T-13910 polymorphism. In this study, we sought to evaluate the clinical value of LCT genotyping in children with GIFD.

Our population had a low prevalence of the LP allele (34.1%), slightly lower than that reported in other Spanish series [10,11,12]. Even though we observed a strong correlation between genotype and lactose malabsorption in children (*p* < 2.2 × 10^−16^; Cramér’s V, 0.54), in line with the findings of studies in adult populations [20,21,22,23,24], we believe that, from a practical perspective, the most important correlation to evaluate is the extent to which genotype is associated with the presence of GI symptoms. In our series, CC carriers were mainly lactose non-absorbers (75.4%), while TT carriers were mainly lactose absorbers (91.6%). Intermediate lactase expression is the phenotype usually associated with CT heterozygosity. Remarkably, in our population, CT individuals were predominantly lactose absorbers (90.7%) and lactose tolerant (74.3%), although we observed a slight increase in the frequency of malabsorption and lactose intolerance in CT children with increasing age. This suggests a decline in lactase activity despite the presence of a T allele.

Despite a global significant correlation between C/T-13910 polymorphism and lactose intolerance (*p* = 2.25 × 10^−09^), our results show that the percentage of intolerance among CC children varies from 8.3% in those ≤5 years to 62.6% in those ≥12 years, suggesting that genetic testing in younger children could not constitute a practical tool for etiological diagnosis of FIGD. Our findings, in agreement with those of Schirru et al. [38], indicate that the clinical value of genetic testing increases with age.

The SNP G/A−22018 is accepted as being in strong linkage disequilibrium with C/T-13190 [39]. Our results support the recommendation that the detection of a single SNP is sufficient for genetic diagnosis of LP in children. In accordance with our C/T-13910 findings, most children with a GG genotype showed lactose malabsorption (90%), while almost all AA children were lactose absorbers.

In our population, peak H_2_ and H_2_ increase were higher in children with lactose intolerance. This may explain the symptoms observed, as the level of gas production has been linked to the presence and severity of intestinal symptoms [40]. Moreover, in lactose intolerant individuals we observed a significant age-associated increase in H_2_ production and a parallel increase in the frequency of abdominal pain. One interesting finding is that time to peak H_2_ was shorter in lactose tolerant than intolerant children. This difference may be linked to an earlier gastric emptying/clearance and shorter intestinal transit times in lactose tolerants, although this hypothesis runs counter to the idea that the rapid movement of hyperosmolar content from the stomach into the intestine causes some of the symptoms identified in lactose intolerant children [41,42], since residual lactase activity is overcome [41,42].

We also evaluated dairy consumption, which is one of several interacting factors that give rise to marked interindividual differences in sensitivity to incompletely absorbed carbohydrates, and that influence the development and severity of symptoms in patients with lactose malabsorption [36]. Another aspect that can influence symptoms is the microbiota, as higher levels of *Bifidobacterium* species have been reported in CC versus TT or CT genotypes in European populations, a difference explained by the greater abundance of lactose available for bacterial fermentation in CC individuals due to lactose malabsorption [43]. Furthermore, probiotic supplementation in lactose-intolerant individuals has beneficial effects on HBT results and on symptoms of lactose intolerance [44].

Contrary to expectations [45,46], we found no significant differences in dairy consumption between lactose tolerant and lactose intolerant children, and no differences in the number of dairy servings per week between genotypes. This finding could be influenced by social, environmental, or behavioral confounding factors, not limited in our study. Even though reduced lactose intake, rather than complete exclusion, is recommended in individuals with lactose intolerance [2], a considerable proportion of our study population did not consume milk or other dairy products daily. This is likely due to several factors, including erroneous attribution of symptoms to lactose intolerance, improper dietary management of lactose intolerance, and the progressive decrease in the consumption of dairy products in the pediatric population in Spain [47,48]. Milk constitutes a basic source of dietary calcium in most Western diets, and dairy consumption in 61% of children evaluated is below the level necessary to achieve the recommended calcium intake [49]. Given that the participants in the present study were evaluated during their growth period, dairy exclusion may have later consequences on phosphocalcic metabolism that were not yet evident in our population, since dairy consumption significantly increases bone mineral content in children [50].

Limitations of our study include methodological limitations inherent to the diagnosis of malabsorption using HBT rather than the gold standard procedure (intestinal biopsy), especially in young children, and the lack of Mendelian randomization in the evaluation of dairy consumption.

## 5. Conclusions

We observed a significant correlation between genotype and phenotype in children with suspected FGID and an increase with age in lactose malabsorption and intolerance (according to HBT results) in LNP subjects. These findings suggest that the practical value of genetic testing is greater in older children.

## Figures and Tables

**Figure 1 nutrients-12-03017-f001:**
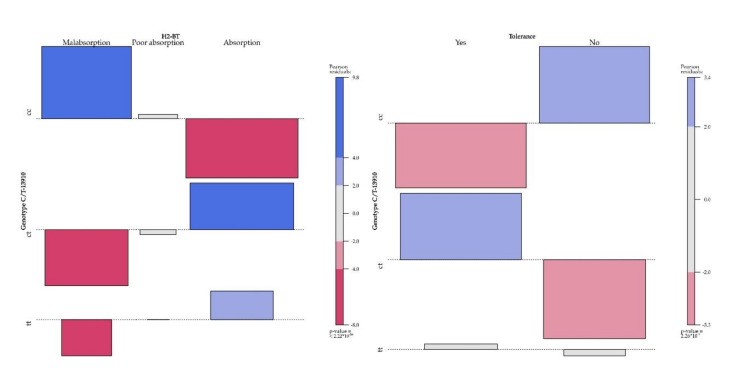
Lactose absorption and tolerance according to C/T-13910 genotype. For the CC genotype, the frequencies of lactose malabsorption and lactose intolerance were higher than expected (positive values shown in blue). Conversely, for the CT genotype, the frequencies of lactose absorption and tolerance were higher than expected.

**Table 1 nutrients-12-03017-t001:** Characteristics of the study population.

	Subjects (*n* = 493)	Age Groups			*p*
		≤5 years (*n* = 50)	6–11 years (*n* = 326)	≥12 years (*n* = 117)	
Age (mean ± SD), y	8.8 ± 3.3	3.4 ± 0.9	7.9 ± 1.9	13.4 ± 1.31	
**Anthropometric characteristics**					
BMI (kg/m^2^)	17.86 ± 3.56	15.51 ± 1.57	17.16 ± 3.01	20.62 ± 3.99	**<2.2 × 10^−16^**
BMI z-score	0.28 ± 1.21	0.12 ± 1.12	0.32 ± 1.23	0.26 ± 1.20	0.446
Underweight	64 [12.9%]	8 [16%]	40 [12.2%]	17 [14.5%]	
Normal weight	297 [60.2%]	32 [64%]	196 [60%]	69 [58.9%]	0.983
Overweight	64 [12.7%]	5 [10%]	44 [13.4%]	15 [12.8%]	
Obesity	68 [13.7%]	5 [10%]	47 [14.4%]	16 [13.6%]	
**Family history of LI**					0.996
Total	52 [10.5%]	1 [2%]	32 [9.8%]	19 [16.2%]	
Father	21 [4.2%]	1 [2%]	15 [4.6%]	5 [4.2%]	
Mother	19 [3.8%]	0	12 [3.6%]	7 [5.9%]	
Sibling	12 [2.4%]	0	5 [1.5%]	7 [5.9%]	
**Previous symptoms**					
Abdominal pain	277 [56.1%]	18 [36%]	194 [59.5%]	65 [55.5%]	**0.008**
Diarrhea	87 [17.6%]	11 [22%]	59 [18%]	17 [14.5%]	0.477
Nausea	34 [6.8%]	0	20 [6.1%]	14 [11.9%]	**0.009**
Vomiting	57 [11.5%]	2 [4%]	43 [13.1%]	12 [10.2%]	0.147
Headache	19 [3.8%]	0	13 [3.9%]	6 [5.1%]	0.313
**C/T-13910 genotype**					
CC	227 [46.04%]	12 [24%]	148 [45.3%]	67 [57.2%]	**0.002**
CT	195 [39.5%]	30 [60%]	127 [38.9%]	38 [32.4%]	
TT	71 [14.4%]	8 [16%]	51 [15.6%]	12 [10.2%]	
C allele frequency	65.80%	54%	64.80%	73.50%	
T allele frequency	34.10%	46%	35.10%	26.40%	
**HBT findings**					
Lactose absorption	280 [56.7%]	42 [84%]	188 [57.6%]	50 [42.7%]	**1.47 × 10^−5^**
Poor lactose absorption	34 [6.8%]	4 [8%]	21 [6.4%]	9 [7.6%]	
Lactose malabsorption	179 [36.3%]	4 [8%]	117 [35.8%]	58 [49.5%]	
Lactose tolerance	288 [58.4%]	36 [72%]	198 [60.7%]	54 [46.1%]	**0.003**
Lactose intolerance	205 [41.5%]	14 [28%]	128 [39.3%]	63 [53.8%]	
Peak H_2_ (ppm)	36.21 ± 48.1	9.56 ± 14.98	35.07 ± 44.1	50.79 ± 61.20	**<0.001**
Time to peak H_2_ (min)	88.72 ± 70.61	71.4 ± 75.18	87.97 ± 72	98.20 ± 63.34	**<0.070**
H_2_ increase (ppm)	31.87 ± 47.56	6.18 ± 14.55	30.66 ± 43.53	46.23 ± 60.81	**<0.001**

BMI: body mass index; H_2_: hydrogen; HBT: hydrogen breath test; LI: lactose intolerance; min: minute; ppm: parts per million; SD: standard deviation; y: years. Data are expressed as the mean ± SD or as number [%]. Captions and significant values are in bold.

**Table 2 nutrients-12-03017-t002:** Comparison of HBT results and induced symptoms in children with lactose malabsorption, tolerance, and intolerance, stratified according to age.

	Malabsorption					Tolerance					Intolerance					*p* ^4^
HBT	Total	≤5 years	6–11 years	≥12 years	*p* ^1^	Total	≤5 years	6–11 years	≥12 years	*p* ^2^	Total	≤5 years	6–11 years	≥12 years	*p* ^3^	
	(*n* = 179)	(*n* = 4)	(*n* = 117)	(*n* = 58)		(*n* = 288)	(*n* = 36)	(*n* = 198)	(*n* = 54)		(*n* = 205)	(*n* = 14)	(*n* = 128)	(*n* = 63)		
**Peak H_2_** (ppm)	87.19 ± 46.9	53.25 ± 18.5	84.73 ± 38.4	94.56 ± 0.9	0.583	23.04 ± 36.5	10.44 ± 15.6	21.02 ± 33.7	38.85 ± 49.5	**0.042**	54.72 ± 55.8	7.28 ± 13.4	56.80 ± 49.3	61.03 ± 68.3	**<0.001**	**<0.001**
**Time to peak H_2_**																
(Mean ± SD, min)	134.41 ± 39.7	165 ± 17.3	135.38 ± 40.4	130.34 ± 38.7	0.736	78.22 ± 72.8	75 ± 75.04	75.90 ± 7	88.88 ± 62.7	**0.005**	103.46 ± 64.7	62.14 ± 77.5	106.64 ± 62.9	106.19 ± 63.2	0.449	**1.02 × 10^−02^**
0 min	0	0	0	0		96 [33.3%]	15 [41.6%]	72 [36.3%]	9 [16.6%]		37 [18%]	8 [57.1%]	19 [14.8%]	10 [15.8%]		
30 min	1 [0.5%]	0	1 [0.8%]	0		40 [13.8%]	2 [5.5%]	29 [14.6%]	9 [16.6%]		12 [5.8%]	0	9 [7%]	3 [4.7%]		
60 min	16 [8.9%]	0	11 [9.4%]	5 [8.7%]		15 [5.2%]	2 [5.5%]	8 [4%]	5 [9.2%]		17 [3.4%]	0	11 [8.5%]	6 [9.5%]		
90 min	27 [15%]	0	16 [13.6%]	11 [18.9%]		19 [6.5%]	2 [5.5%]	12 [6%]	5 [9.2%]		25 [12.1%]	1 [7.1%]	15 [11.7%]	9 [14.2%]		
120 min	40 [22.3%]	0	24 [20.5%]	16 [27.5%]		27 [9.3%]	4 [11.1%]	10 [5%]	13 [24%]		29 [14.1%]	1 [7.1%]	20 [15.6%]	8 [12.6%]		
150 min	43 [24%]	2 [50%]	29 [24.7%]	11 [18.9%]		30 [10.4%]	4 [11.1%]	22 [11.1%]	4 [7.4%]		40 [19.5%]	2 [14.2%]	25 [19.5%]	13 [20.6%]		
180 min	53 [29.6%]	2 [50%]	36 [30.7%]	15 [25.9%]		61 [21.1%]	7 [19.4%]	45 [22.7%]	9 [16.6%]		45 [21.9%]	2 [14.2%]	29 [22.6%]	14 [22.2%]		
**H_2_ increase**																
(Mean ± SD, ppm)	82.64 ± 46.3	50.05 ± 19	80.29 ± 37.4	89.60 ± 60.9	0.623	18.85 ± 35.7	6.88 ± 15.6	16.91 ± 32.9	33.92 ± 48.7	**0.025**	50.17 ± 55.1	4.35 ± 11.6	51.92 ± 49.1	56.79 ± 68.1	**<0.001**	**<0.001**
≤10 ppm	0	0	0	0		202 [70%]	30 [83.3%]	144 [73%]	28 [51.8%]		80 [39%]	13 [92.8%]	45 [35.1%]	22 [34.9%]		
10–20 pm	19 [10.6%]	0	0	0		23 [7.9%]	3 [8.3%]	14 [7%]	6 [11.1%]		13 [6.3%]	0	7 [5.4%]	5 [7.9%]		
20–30 pm	14 [7.8%]	0	14 [11.9%]	5 [8.7%]		7 [2.4%]	0	6 [3%]	1 [1.8%]		8 [3.9%]	0	5 [3.9%]	2 [3.1%]		
30–40 pm	10 [5.5%]	1 [25%]	10 [8.5%]	5 [8.7%]		7 [2.4%]	1 [2.7%]	4 [2%]	2 [3.7%]		7 [3.4%]	0	5 [3.9%]	3 [4.7%]		
40–50 ppm	14 [7.8%]	2 [50%]	2 [1.7%]	6 [10.3%]		4 [1.3%]	1 [2.7%]	1 [0.5%]	2 [3.7%]		6 [2.9%]	1 [7.1%]	1 [0.7%]	4 [6.3%]		
50–100 ppm	80 [44.6%]	1 [25%]	63 [53.8%]	29 [50%]		32 [11.1%]	1 [2.7%]	19 [9.5%]	12 [22.2%]		62 [30.2%]	0	44 [34.3%]	18 [28.5%]		
>100 ppm	42 [23.4%]	0	30 [25.6%]	13 [22.1%]		12 [4.1%]	0	10 [5%]	3 [5.5%]		29 [14.1%]	0	20 [15.6%]	9 [14.2%]		
**Symptoms**																
Abdominal pain	96 [53.6%]	1 [25%]	59 [50.4%]	36 [62%]	0.196	—	—	—	—		165 [80.4%]	10 [71.4%]	101 [78.9%]	54 [85.7%]	0.340	
Flatulence	28 [15.6%]	1 [25%]	22 [18.8%]	5 [8.7%]	0.125	—	—	—	—		58 [28.2%]	6 [42.8%]	35 [27.3%]	17 [26.9%]	0.455	
Diarrhea	36 [20.1%]	4 [100%]	28 [23.9%]	8 [13.7%]	0.173	—	—	—	—		49 [23.9%]	3 [21.4%]	35 [27.3%]	11 [17.4%]	0.308	
Nausea	16 [8.9%]	1 [25%]	10 [8.5%]	5 [8.7%]	1	—	—	—	—		4 [13.6%]	1 [7.1%]	15 [11.7%]	12 [19%]	0.359	
Vomiting	3 [16.7%]	4 [100%]	2 [1.7%]	1 [1.7%]	1	—	—	—	—		4 [1.9%]	0	3 [2.3%]	1 [1.5%]	1	
**Symptoms (n)**																
Mean ± SD	1.03 ± 0.9	0.75 ± 1.5	1.08 ± 1.03	0.96 ± 0.8	0.651	—					1.52 ± 0.6	1.42 ± 0.7	1.53 ± 0.7	1.53 ± 0.59	0.593	
1 symptom	61 [34%]	1 [25%]	39 [33.3%]	22 [37.9%]		—	—	—	—		117 [57%]	10 [71.4%]	75 [58.5%]	32 [50.7%]	**<0.001**	
2 symptoms	39 [21.7%]	0	25 [21.3%]	14 [24.1%]		—	—	—	—		70 [59.8%]	2 [14.2%]	40 [31.2%]	28 [44.4%]	**1.02 × 10^+03^**	
3 symptoms	14 [7.8%]	3 [75%]	11 [9.4%]	2 [3.4%]		—	—	—	—		17 [8.2%]	2 [14.2%]	12 [9.3%]	3 [4.7%]	**<0.001**	
≥3 symptoms	1 [0.5%]	0	1 [0.8%]	0			—	—	—		1 [0.4%]	0	1 [0.7%]	0		

HBT: hydrogen breath test; min: minutes; n: number; —: none; ppm: parts per million; SD: standard deviation. Malabsorption is defined as an expired H_2_ increase ≥20 ppm with respect to baseline, and intolerance as the presence of symptoms during the HBT test. *p*^1^, comparison between age groups among children with lactose malabsorption; *p*^2^, comparison between age groups among children with lactose tolerance; *p*^3^, comparison between age groups among children with lactose intolerance; *p*^4^, comparison between lactose-tolerant and lactose-intolerant groups. For the malabsorption group, comparisons were made only for children ≤12 years and >12 years, as only 4 children ≤5 years presented malabsorption. Captions and significant values are in bold.

**Table 3 nutrients-12-03017-t003:** C/T-13190 polymorphism and HBT according to age. Genotype-phenotype correlation.

HBT	CT-13910 Polymorphism															*p* ^4^
	CC					CT					TT					
	Total	≤5 years	6−11 years	≥12 years	*p* ^1^	Total	≤5 years	6−11 years	≥12 years	*p* ^2^	Total	≤5 years	6−11 years	≥12 years	*p* ^3^	
	(*n* = 227)	(*n* = 12)	(*n* = 148)	(*n* = 67)		(*n* = 195)	(*n* = 30)	(*n* = 127)	(*n* = 38)		(*n* = 71)	(*n* = 8)	(*n* = 51)	(*n* = 12)		
Absorption	38 [16.7%]	7 [58.3%]	24 [16.2%]	7 [10.4%]		177 [90.7%]	28 [93.3%]	116 [91.3%]	33 [86.8%]		65 [91.6%]	7 [87.5%]	48 [94.1%]	10 [83.3%]		<2.22 × 10^−16^
Poor absorption	18 [7.9%]	2 [16.6%]	12 [8.1%]	4 [5.9%]	0.001	11 [5.6%]	1 [3.3%]	7 [5.5%]	3 [7.8%]	0.850	5 [7%]	1 [12.5%]	2 [3.9%]	2 [16.6%]	0.412	
Malabsorption	171 [75.4%]	3 [25%]	112 [75.6%]	56 [83.5%]		7 [3.7%]	1 [3.3%]	4 [3.1%]	2 [5.2%]		1 [1.4%]	0	1 [1.9%]	0		
Tolerance	100 [44%]	11 [91.6%]	64 [43.4%]	25 [37.3%]		145 [74.3}	20 [66.6%]	103 [81.1%]	22 [57.8%]		43 [60.5%]	5 [62.5%]	31 [60.7%]	7 [58.3%]	1	2.26 × 10^−09^
Intolerance	127 [56%]	1 [8.3%]	84 [56.7%]	42 [62.6%]	0.002	50 [25.7%]	10 [33.3%]	24 [18.8%]	16 [42.1%]	**0.009**	28 [39.5%]	3 [37.5%]	20 [39.2%]	5 [41.6%]		

HBT: hydrogen breath test. Malabsorption is defined as an expired H_2_ increase ≥20 ppm with respect to baseline, and intolerance as the presence of symptoms during the HBT test. *p*^1^, comparison between age groups among CC subjects. *p*^2^, comparison between age groups among CT subjects; *p*^3^, comparison between age groups among TT subjects; *p*^4^, correlation between phenotype by HBT and C/T-13910 genotype. Percentages are expressed in brackets.

**Table 4 nutrients-12-03017-t004:** Dairy product consumption and phosphocalcic metabolism.

	Total (*n* = 120)	Age Groups			*p* ^1^	Tolerance (*n* = 56)	Intolerance (*n* = 64)	*p* ^2^
		≤5 years (*n* = 6)	6–11 years (*n* = 71)	≥12 years (*n* = 43)				
G/A-22018 polymorphism					**<2.2 × 10^−16^**			**0.002**
GG	84 [70%]	1 [16.6%]	47 [66.1%]	36 [83.7%]		31 [55.3%]	53 [82.8%]	
GA	29 [24.1%]	4 [66.6%]	19 [26.7%]	6 [13.9%]		21 [37.5%]	8 [12.5%]	
AA	7 [5.8%]	1 [16.6%]	5 [7%]	1 [2.3%]		4 [7.1%]	3 [4.6%]	
G allele frequency [%]	82%	50%	79.50%	90.60%		74.10%	89%	
A allele frequency [%]	18%	50%	20.50%	9.40%		25.90%	11%	
Number of dairy servings/week								
Mean ± SD	12.89 ± 4.89	11.63 ± 3.14	12.87 ± 5.13	12.25 ± 4.71	0.46	13.35 ± 4.92	11.92 ± 4.80	0.165
<7	10 [8.3%]	0	6 [8.4%]	5 [11.6%]		3 [5.36%]	8 [12.5%]	
7	8 [6.6%]	1 [16.6%]	5 [7%]	2 [4.6%]		4 [7.14%]	4 [6.25%]	
7–10	16 [13.3%]	3 [50%]	6 [8.4%]	9 [20.9%]		7 [12.5%]	9 [14.07%]	
11–14	40 [33.3%]	2 [33.3%]	25 [35.2%]	12 [27.9%]		20 [35.71%]	20 [31.26%]	
14–17	31 [25.8%]	0	21 [29.5%]	9 [20.9%]		11 [19.64%]	22 [34.38%]	
>17	14 [11.6%]	0	8 [11.2%]	6 [13.9%]		11 [19.64%]	1 [1.56%]	
Number of milk servings/week								
0	13 [10.8%]	0	9 [12.6%]	4 [9.4%]	0.643	5 [8.93%]	8 [12.5%]	0.501
1–2	6 [5%]	0	5 [7.2%]	1 [2.3%]		3 [5.36%]	3 [4.69%]	
3–4	7 [5.8%]	0	1 [1.4%]	6 [13.9%]		3 [5.36%]	4 [6.26%]	
5–7	94 [78.4%]	6 [100%]	56 [78.8%]	32 [74.4%]		45 [80.36%]	49 [76.56%]	
Number of yogurt servings/week								
0	19 [15.8%]	1 [16.6%]	11 [15.4%]	7 [16.2%]	0.213	8 [14.29%]	11 [17.19%]	0.238
1–2	15 [12.6%]	0	8 [11.3%]	7 [16.2%]		5 [8.93%]	10 [15.63%]	
3–4	30 [25%]	3 [50%]	15 [21.1%]	12 [27.9%]		15 [26.78%]	15 [23.44%]	
5–7	56 [46.6%]	2 [33.3%]	37 [52.2%]	17 [39.5%]		28 [50%]	28 [43.75%]	
Number of cheese servings/week								
0	34 [28.3%]	4 [66.6%]	18 [25.4%]	12 [28.1%]	0.872	17 [30.36%]	17 [26.56%]	0.317
1–2	37 [30.8%]	2 [33.6%]	24 [33.8%]	11 [25.5%]		12 [21.43%]	25 [39.06%]	
3–4	23 [19.1%]	0	12 [16.9%]	11 [25.5%]		13 [23.22%]	10 [15.63%]	
5–7	26 [21.6%]	0	17 [23.9%]	9 [20.9%]		14 [25%]	12 [18.75%]	
Estimated weekly milk intake								
Mean ± SD	2604.58 ± 1477.60	2154.16 ± 579.56	2585.71 ± 1488.12	2698.25 ± 1554.54	0.972	2771.42 ± 1545.62	2458.51 ± 1411.40	0.459
≤500 mL	10 [8.3%]	0	7 [9.8%]	3 [6.9%]		3 [5.36%]	7 [10.93%]	
500–1000 mL	10 [8.3%]	0	6 [8.4%]	4 [9.3%]		6 [10.72%]	4 [6.25%]	
1000–2000 mL	23 [19.1%]	1 [16.6%]	12 [16.9%]	10 [23.2%]		9 [16.08%]	14 [21.87%]	
2000–3000 mL	36 [30%]	5 [83.3%]	22 [30.9%]	9 [20.9%]		16 [25.59%]	20 [31.24%]	
3000–4000 mL	21 [17.5%]	0	12 [16.9%]	9 [20.9%]		9 [16.08%]	12 [18.75%]	
4000–5000 mL	11 [9.1%]	0	8 [11.2%]	3 [6.9%]		7 [12.53%]	4 [6.26%]	
>5000 mL	9 [7.5%]	0	4 [5.6%]	5 [11.6%]		6 [10.72%]	3 [4.68%]	
Phosphocalcic metabolism								
Calcium (mg/dL)	9.79 ± 0.28	9.73 ± 0.23	9.82 ± 0.28	9.74 ± 0.27	0.401	9.81 ± 0.26	9.77 ± 0.29	0.308
Phosphorous (mg/dL)	4.77 ± 0.55	4.93 ± 0.46	4.88 ± 0.41	4.56 ± 0.70	**0.001**	4.82 ± 0.54	4.72 ± 0.55	0.317
PTH (pg/mL)	40.09 ± 16.64	39.83 ± 21.02	37.11 ± 14.03	45.04 ± 21.02	0.086	38.79 ± 14.38	41.23 ± 18.44	0.596
25-OH vitamin D (ng/mL)	15.11 ± 6.24	15.16 ± 6.30	15.35 ± 6.70	14.72 ± 5.51	0.545	15.39 ± 6.21	14.87 ± 6.30	0.954
Calcitriol (pg/mL)	56.15 ± 15.60	61 ± 14.54	53.88 ± 13.03	61 ± 14.54	0.167	60.12 ± 16.53	52.67 ± 13.96	**0.011**

PTH: parathormone. Lactose tolerance is defined as the absence of symptoms during the HBT test and intolerance as the presence of symptoms during it. *p*^1^, comparison between age groups; *p*^2^, comparison of lactose tolerant versus lactose intolerant subjects. Percentages are expressed in brackets. Significant values are in bold.

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
