# Peer review of "Clinical Utility of LCT Genotyping in Children with Suspected Functional Gastrointestinal Disorder"

_nutrients, 2020, doi:10.3390/nu12103017_

Round 1

Reviewer 1 Report

As above:

This is a very interesting study indeed.
All goes almost well until the conclusion which does not add up with the results.
In conclusion it is stated:
1- "assessment of lactose malabsorption is of little utility in the study of GIFD in younger children"
- From this study, I understood that there was a strong correlation between the genetic testing and "malabsorption" assessment which of course is not clear what is the nature of this assessment. This part of the conclusion to me contradicts with the finding in the results and needs further clarification. Why this assessment is of little value in younger children?

2- The 2nd part of the conclusion states: "and the clinical utility of genetic testing in this population increases in children >5 years."
- Similar to the above, there was a strong agreement between with two methods and here suddenly it is decided that genetic testing is better in this age group.
- Also, I feel, "utility" here means the clinical "value" rather than the amount of use in the clinics (?).

Other comments:
- one important clarification can be that what is defined as lactose malabsorption/intolerance? is it based on the symptoms after ingesting dairy products or based on Hydrogen breath testing or else?

- Also, was there any one to one correlation between the genotyping and malabsorption/intolerance? How many of those who did have malabsorption also had positive genotyping result? In the current text, it seems that eg. 36% study population had malabsorption and 46% had CC positive but may be those 36 and these 46 are not the same patient (concordance report)?

Author Response

REVIEWER 1

This is a very interesting study indeed. All goes almost well until the conclusion which does not add up with the results.

In conclusion it is stated:

1.- "Assessment of lactose malabsorption is of little utility in the study of GIFD in younger children". From this study, I understood that there was a strong correlation between the genetic testing and "malabsorption" assessment which of course is not clear what is the nature of this assessment. This part of the conclusion to me contradicts with the finding in the results and needs further clarification. Why this assessment is of little value in younger children?

Answer: As the reviewer remarks, we observed a significant correlation between hypolactasia genotype and malabsorption. Moreover, this correlation increased with age. Table 3 shows how increasing age is associated with a progressive increase in the percentage of malabsorbers among CC subject. Nonetheless, in line with the reviewer’s comment we have altered the conclusion (see below) to better reflect our findings and avoid confusion.

Line 370. “We observed a significant correlation between genotype and phenotype in children with suspected GIFD and an increase with age in lactose malabsorption and intolerance (according to HBT results) in LNP subjects. These findings suggest that the practical value of genetic testing is greater in older children”.

We have also modified the conclusion in the abstract (line 30):

   “In conclusion, we found a significant correlation between genotype and phenotype, greater in older children, suggesting that the clinical value of genetic testing increases with age”.

2.- The 2nd part of the conclusion states: "and the clinical utility of genetic testing in this population increases in children >5 years."

- Similar to the above, there was a strong agreement between with two methods and here suddenly it is decided that genetic testing is better in this age group.

- Also, I feel, "utility" here means the clinical "value" rather than the amount of use in the clinics (?).

Answer: As described above, our findings suggest that genetic testing to identify lactose malabsorption as a cause of symptoms compatible with suspected gastrointestinal functional disorders is of greater value in older children, since the percentage of lactose malabsorbers among CC individuals increased from 25% in children ≤5 years to 83.5% in those >12 years (Table 3). As indicated in the previous response, we have edited the manuscript for greater clarity on this point. We thank the reviewer for their suggestion to substitute “utility” with “value” and have applied this change in the abstract and the conclusions section.

Other comments:

  1. - One important clarification can be that what is defined as lactose malabsorption/intolerance? is it based on the symptoms after ingesting dairy products or based on Hydrogen breath testing or else?

Answer: Malabsorption was defined according to HBT test results as an H2 increase ≥20 ppm with respect to baseline, and intolerance was defined as the presence of symptoms during the HBT test.

To clarify this point, we have included the following in the Material and Methods section:

   Line 104. “Malabsorption was defined according to HBT test results as a H2 increase ≥20 ppm with respect to baseline, and intolerance was defined as the presence of symptoms during the HBT test”.

We have also included this information in the description of the study objectives:

   Lines 85-86. “In this study we assessed the role of lactose malabsorption in gastrointestinal functional disorders (GIFD) in children, and examined the correlation between the LNP genotype and phenotype, defined based on exhaled hydrogen (H2) measurement and gastrointestinal (GI) symptomatology during HBT, in a pediatric population”.

4 - Also, was there any one to one correlation between the genotyping and malabsorption/intolerance? How many of those who did have malabsorption also had positive genotyping result? In the current text, it seems that eg. 36% study population had malabsorption and 46% had CC positive but may be those 36 and these 46 are not the same patient (concordance report)?

Answer: The percentage of CC subjects in our study population was 46.04%. As mentioned above, Table 3 shows the number of CC subjects who were lactose absorbers (16.7%), poor absorbers (7.9%), and malabsorbers (75.4%). The same table shows that of the 179 children with lactose malabsorption, 171 were CC (95.5%), 7 were CT (3.9%), and 1 was TT (0.5%) (p <2.2e-16)).

On reflection, we feel that Table 3 was difficult to read and we have since edited it for greater ease of interpretation.

Reviewer 2 Report

The paper by Couce et al. describes a retrospective cohort of 493 children with suspected lactose intolerance who underwent both genetic and breath testing. 

The paper is nicely written and addresses a neglected field of pediatric gastroenterology.

Few modifications are required: 

1) please substitute "symptomatology" with symptoms throughout the manuscript and tables.

2) please change sex with gender throughout the manuscript and tables.

3) please change vomits with vomiting throughout the manuscript and tables.

4) please change cephalea with headache throughout the manuscript and tables

5) PLEASE FURTHER COMMENT IN THE METHODS THE INTERPRETATION OF COHEN'S K. TO MY KNOWLEDGE K -0.36 DOES NOT MEAN AGREEMENT AT ALL

6) Discussion part:

PLEASE FEW WORDS ON THE NEED TO EXCLUDE CELIAC DISEASE IN THESE CHILDREN. TO THIS REGARD PLEASE CITE BOTH THE FOLLOWING PAPERS: a) PMID: 31076989; b) PMID: 24756157

Author Response

REVIEWER 2

The paper by Couce et al. describes a retrospective cohort of 493 children with suspected lactose intolerance who underwent both genetic and breath testing.The paper is nicely written and addresses a neglected field of pediatric gastroenterology.

Few modifications are required:

  • Please substitute "symptomatology" with symptoms throughout the manuscript and tables.Please change sex with gender throughout the manuscript and tables. Please change vomits with vomiting throughout the manuscript and tables. Please change cephalea with headache throughout the manuscript and tablesPlease

Answer: According with the recommendations we have made the style modifications suggested

  • Please further comment in the Methods the interpretation of Cohen´s K. To my Knowledge K -0.36 does not mean agreement at all.

Answer:  We thank the reviewer comment as it allow us to detect a mistake in the manuscript as the Cohen´s Kappa is 0.55 ( 95% CI, 0.49–0.61). corresponding to moderate agreement.

We change this data in line 221.

  • ) Discussion part:

Please few words on the need to exclude celiac disease in these children to this regard. Plase cite both the following papers:: a) PMID: 31076989; b) PMID: 24756157

Answer: We have included the next paragraph in the Discussion section:

“When evaluating GIFD in infants and children, we should screen for celiac disease (CD) as its prevalence among children with irritable bowel syndrome  is higher than in general pediatric population [32]. It should be borne in mind that allergy to cow’s milk protein  and CD may act as a predisposing or coexisting factor, potentially contributing to inflammation and visceral hypersensitivity in early life, which may manifest as GIFD [33, 34].”

And we add the suggested cites to Bibliography:

32-Cristofori F, Fontana C, Magistà A, Capriati T, Indrio F, Castellaneta S, Cavallo L, Francavilla R. Increased prevalence of celiac disease among pediatric patients with irritable bowel syndrome: a 6-year prospective cohort study. JAMA Pediatr. 2014;168(6):555-60. doi: 10.1001/jamapediatrics.2013.4984.

34- Valitutti F, Fasano A. Breaking Down Barriers: How Understanding Celiac Disease Pathogenesis Informed the Development of Novel Treatments. Dig Dis Sci. 2019 Jul;64(7):1748-1758. doi: 10.1007/s10620-019-05646-y. PMID: 31076989; PMCID: PMC6586517.

Reviewer 3 Report

The topic of the manuscript is actual and very important, as functional gastrointestinal disorders are very common. However, I have some methodological concerns, as following:

  • Patients with GI  symptomatology  compatible with GIFD were evaluated (lines 82-83). The authors should specify how GIFD were defined; in other words, which criteria were used to define GIFD (Rome criteria? Anything else?). It is not clear, also seeing symptoms in Table 1 (e.g cefalea)
  • It is well known that LNP genotype does not mean “lactose intolerance”; furthermore the clinical utility of breath test is very discussed, also in patients older than 5 years. Throughout the text the difference between genotype predisponing to malabsorption and lactose intolerance (e.g clinical symptoms), and also the Table 4 is quite confusing: it is not clear what lactose tolerance and intolerance stand for (just based on HBT results?). Table 3 is very difficult to read: what does correlation between lactose malabsorption and lactose intolerance according C/T-13910 genotype exactly mean?
  • The authors should better clarify the difference between clinical symptoms of lactose intolerance and role of breath test, e.g lactose malabsorption, critically addressing utility and limits in the Discussion
  • The Discussion section should be partly rewritten, clearly focusing on the observed results, better addressing the limitations of the present study
  • In the discussion section, the authors state that recurrent abdominal pain due to a suspected functional disorder, lactose malabsorption appears not to play a major role, particularly in younger children. I agree with this statement. I invite the authors to consider that cow’s milk allergy may play an important role as predisposing or coexisting factor in GIFD. Thus, the importance of a correct classification and definition of GIFD is crucial; this possible relationship  should be considered and briefly discussed

Author Response

REVIEWER 3

REVIEWER 3

The topic of the manuscript is actual and very important, as functional gastrointestinal disorders are very common. However, I have some methodological concerns, as following:

1 - Patients with GI symptomatology compatible with GIFD were evaluated (lines 82-83). The authors should specify how GIFD were defined; in other words, which criteria were used to define GIFD (Rome criteria? Anything else?). It is not clear, also seeing symptoms in Table 1 (e.g cefalea).

Answer: We defined GIFD in accordance with Rome IV criteria. This information has been included in the abstract and in the following sentence in the Study design and population section:

   Line 95-97. “Between January 1st, 2010, and December 31st, 2019, 493 consecutive pediatric patients (sample A) with GI symptomatology compatible with GIFD, defined according Rome IV criteria [27], were evaluated”.

We have also added a new citation: [27]: Hyams, J.S.; Di Lorenzo, C.; Spas, M.; Shulman, R.J.; Staiano, A.; van Tilburg, M. Functional Disorders: Children and Adolescent. Gastroenterology 2016, S0016-5085(16)00181-5. doi: 10.1053/j.gastro.2016.02.015.

We also agree with the reviewer than the inclusion of “cephalea” may be confusing and we have removed this term from Table 2.

2 - It is well known that LNP genotype does not mean “lactose intolerance”; furthermore the clinical utility of breath test is very discussed, also in patients older than 5 years. Throughout the text the difference between genotype predisponing to malabsorption and lactose intolerance (e.g clinical symptoms), and also the Table 4 is quite confusing: it is not clear what lactose tolerance and intolerance stand for (just based on HBT results?). Table 3 is very difficult to read: what does correlation between lactose malabsorption and lactose intolerance according C/T-13910 genotype exactly mean?

Answer: As described in section 2.4 of article, we define lactose tolerance as the absence of symptoms during the HBT test and intolerance as the presence of symptoms during the test, regardless of exhaled H2 measurement.

We have clarified in the text that phenotype in our study was defined based on HBT results. For greater clarity we have added the following sentence in the legend of Table 4:

Lactose tolerance is defined as the absence of symptoms during the HBT test and intolerance as the presence of symptoms during it”.

We agree with the reviewer on the difficulty in interpreting the data presented in Table 3, and have modified the table for greater ease of interpretation. Specifically, we have altered the wording of the legend in Table 3, as follows:

   “p4, correlation between phenotype by HBT and C/T-13910 genotype, and between lactose                       tolerance/intolerance diagnosis and C/T-13910 genotype”.

We have also included the following sentence in the legends accompanying Tables 2 and 3: “Malabsorption is defined as an expired H2 increase ≥20 ppm with respect to baseline, and intolerance as the presence of symptoms during the HBT test” .

3- The authors should better clarify the difference between clinical symptoms of lactose intolerance and role of breath test, e.g lactose malabsorption, critically addressing utility and limits in the Discussion. The Discussion section should be partly rewritten, clearly focusing on the observed results, better addressing the limitations of the present study.

Answer: In line with the reviewer’s suggestions, we have edited the discussion to better describe the results and highlight the limitations of the study.

4- In the discussion section, the authors state that recurrent abdominal pain due to a suspected functional disorder, lactose malabsorption appears not to play a major role, particularly in younger children. I agree with this statement. I invite the authors to consider that cow’s milk allergy may play an important role as predisposing or coexisting factor in GIFD. Thus, the importance of a correct classification and definition of GIFD is crucial; this possible relationship should be considered and briefly discussed.

Answer: We thank the reviewer for their suggestion, based on which we have included the following paragraph in the Discussion section:

Line 285-289. “When evaluating GIFD in infants and children, it should be borne in mind that allergy to cow’s milk protein may act as a predisposing or coexisting factor, potentially contributing to inflammation and visceral hypersensitivity in early life, which may manifest as GIFD [32]”.

The following additional reference has been added to the bibliography: [32]. Pensabene,L.; Salvatore, S.; D´Auria, E.; Parisi, F.; Concolino, D.; Borreli, O.; et al. Cow’s Milk Protein Allergy in Infancy: A Risk Factor for Functional Gastrointestinal Disorders in Children?. Nutrients, 2018, 10, 1716. doi:10.3390/nu10111716.